# Thwarting Adversarial Examples: An $L_0$-Robust Sparse Fourier Transform

**Mitali Bafna** [*]
School of Engineering & Applied Sciences
Harvard University
Cambridge, MA USA
`mitalibafna@g.harvard.edu`

**Jack Murtagh** [*]
School of Engineering & Applied Sciences
Harvard University
Cambridge, MA USA
`jmurtagh@g.harvard.edu`

**Nikhil Vyas**[*]
Department of Electrical Engineering and Computer Science
MIT
Cambridge, MA USA
`nikhilv@mit.edu`

## Abstract

We give a new algorithm for approximating the Discrete Fourier transform of an approximately sparse signal that has been corrupted by worst-case $L_0$ noise, namely a bounded number of coordinates of the signal have been corrupted arbitrarily. Our techniques generalize to a wide range of linear transformations that are used in data analysis such as the Discrete Cosine and Sine transforms, the Hadamard transform, and their high-dimensional analogs. We use our algorithm to successfully defend against well known $L_0$ adversaries in the setting of image classification. We give experimental results on the Jacobian-based Saliency Map Attack (JSMA) and the Carlini Wagner (CW) $L_0$ attack on the MNIST and Fashion-MNIST datasets as well as the Adversarial Patch on the ImageNet dataset.

## 1   Introduction

In the last several years, neural networks have made unprecedented achievements on computational learning tasks like image classification. Despite their remarkable success, neural networks have been shown to be brittle in the presence of adversarial noise [SZS+13]. Many effective attacks have been proposed in the context of computer vision that reliably generate small perturbations to input images (sometimes imperceptible to humans) that drastically change the network's classification of the image [MFF15, GSS15, CW16]. As deep learning becomes more integrated into our everyday technology, the need for systems that are robust to adversarial noise grows, especially in applications to security.

A lot of work has been done to improve robustness and defend against adversarial attacks [PMW+16, TKP+17, MMS+17]. However many approaches rely on knowing the attack strategy in advance and too few proposed methods for robustness offer theoretical guarantees and may be broken by a new attack shortly after they're published. As such, recent deep learning literature has seen an arms race of back-and-forth attacks and defenses reminiscent of cryptography before it was grounded in firm theoretical foundations.

In this work, we give a framework for improving the robustness of classifiers to adversaries with $L_0$ *noise budgets*. That is, adversaries are restricted in the number of features they can corrupt, but may corrupt each arbitrarily. Our framework is based on a new Sparse Discrete Fourier transform (DFT) that is robust to worst-case $L_0$ noise added in the time domain. We call such transformations

---

[*]Authors ordered alphabetically.

$L_0$-*robust*. In particular, we show how to recover top coefficients of an approximately sparse signal that has been corrupted by worst-case $L_0$ noise.

Our theoretical results use techniques from compressed sensing [CRT06, BCDH10]. In fact we provide a much more general framework for building $L_0$-robust sparse transformations that applies to many transformations used in practice such as all discrete variants of the Fourier transform, the Sine and Cosine Transforms, the Hadamard transform, and their higher-dimensional generalizations. Our approach can be used to develop algorithms with the following benefits:

- **Provable performance guarantees.** Our approach leverages rigorous results in compressed sensing that allow us to prove theorems about $L_0$-robust sparse transformations under mild assumptions that typically hold in practice.

- **Worst-case adversaries.** The guarantees of our algorithms hold for all adversaries that stay within the noise budget, given that the input signals are sparse in the Fourier or related domains. In particular, our defenses do not require prior knowledge of the adversary's attack strategy. In Section 4.2, we relax this and show that one can design improved algorithms within our framework when given more knowledge of the adversary.

- **Generality.** Our framework is general purpose and is compatible with a variety of basis transformations commonly used in scientific computing.

A notable feature of our framework is the focus on $L_0$ noise. This threat model has been considered in previous works and $L_0$ attacks and defenses have been developed [CW16, PMJ$^+$15, PMW$^+$16]. While $L_2$ attacks are more commonly studied, many of the most high-profile recent real-world attacks actually fit in the $L_0$ model, such as the graffiti-like road sign perturbations of [EEF$^+$17], the eyeglasses that fool facial recognition software [SBBR16], and the patch that can make almost any image get labeled as a 'toaster' by state-of-the-art classifiers [BMR$^+$17]. In general, physical obstructions in images or malicious splicing of audio or video files are realistic threats that can be modeled as $L_0$ noise, whereas $L_2$ attacks may be more difficult to carry out in the physical world.

The connection between our $L_0$-robust transformations and adversarial attacks on images is as follows. Many natural images are sparse in Fourier bases such as the basis used in the Discrete Cosine transform (DCT). Indeed, this is a necessary feature for compression algorithms like JPEG to work. Through this lens, corrupted images can be viewed as noisy signals that are sparse in some domain and our techniques allow us to reconstruct these sparse signals under worst-case/adversarial $L_0$ noise. This reconstruction allows us to correct the corruptions made by the adversary to produce something close to the original image, which in turn improves the neural network accuracy.

Our results have wide applicability in signal processing since it is well known that audio/video signals are sparse in the Fourier or wavelet domains. Signal processing is important in many areas of science and medicine including MRI, radio astronomy, and facial recognition. Errors are ubiquitous in the above applications, whether due to natural artifacts, sensor failures, or malicious tampering.

In Section 4, we give experimental results that demonstrate the effectiveness of our approach against leading $L_0$ attacks. For example, in one experiment the network accuracy drops from $88.5\%$ on uncorrupted images to $24.8\%$ on adversarial images with 30 pixels corrupted, but after our correction, network accuracy returns to $83.1\%$. On another attack, the adversary is free to choose its own budget and network accuracy drops from $87.8\%$ all the way to $0\%$ (the adversary succeeds on every image) but after running our correction algorithm, network accuracy returns to $85.7\%$.

In Section 2, we set up the problem and discuss related work. We give new theoretical results in Section 3. In Section 4, we evaluate our framework on three leading $L_0$ attacks in the literature: the JSMA attack of Papernot et al [PMJ$^+$15], the $L_0$ attack from Carlini and Wagner (CW) [CW16], and the adversarial patch from Brown et al [BMR$^+$17].

**Notation**   For a vector $v$, we let $v_{h(k)}$ and $v_{t(k)}$ denote the head($k$) and tail($k$) of $v$. That is, $v_{h(k)}$ denotes the vector containing just the $k$ largest coordinates of $v$ in absolute value with all other coordinates set to 0 and $v_{t(k)} = v - v_{h(k)}$. For example if $v = [-3, 2, 1]$ then $v_{h(2)} = [-3, 2, 0]$ and $v_{t(2)} = [0, 0, 1]$. We refer to $v_{h(k)}$ as the "top $k$ coefficients of $v$". For a vector $v$, we let $\hat{v} = Fv$, where $F$ is the contextual linear transformation. We use the phrase, "projection of $v$ to its top-$k$ $F$-coefficients", to mean the result of $F^{-1}(Fv)_{h(k)}$. That is, calculate the top-$k$ coefficients of $Fv$, set the remaining coefficients to 0 and then invert the result to back to the original domain by applying

$F^{-1}$. Unless specified, $\|\cdot\|$ denotes the $L_2$ norm of a vector. For a scalar $c$, $|c|$ denotes the absolute value when $c \in \mathbb{R}$ and denotes the modulus of $c$ when $c \in \mathbb{C}$. We say that a vector $v$ is $k$-sparse if all but $k$ of its entries are 0. We say that $v$ is approximately $(k, \epsilon)$-sparse if $\|x_{t(k)}\| \leq \epsilon \cdot \|x\|$. We define $\mathcal{M}_k$ to be the set of all $k$-sparse vectors. $\vec{0}$ denotes the all-zeroes vector.

## 2 Overview

### 2.1 $L_0$-Robust Sparse Fourier Transform

**Problem Setup:** The key property that we use is that natural images are approximately sparse in frequency bases like the 2D Discrete Fourier basis or the 2D Discrete Cosine basis. This sparsity is exploited in image and video compression algorithms like JPEG and MPEG. The DFT and DCT are just linear transformations (in fact change of bases) from the space of images to a frequency domain. So given a $d \times d$ image, we model it as approximately sparse in one of these bases, which from now on we will just refer to as the 'Fourier basis'. Note that once the basis is fixed we can think of the image $x \in \mathbb{R}^n$ ($n = d^2$) as an approximately sparse vector in the corresponding Fourier basis.

Our goal is to approximate the top-$k$ Fourier coefficients of a vector $x$ even after it has been corrupted with adversarial $L_0$ noise. We do not know the locations or magnitudes of the corruptions but we do assume that we know an upper bound on the number of corrupted coordinates. In other words, if $F$ is the Fourier matrix (the matrix corresponding to the Discrete Fourier linear transformation), we want to approximate $\hat{x}_{h(k)}$ where $\hat{x} = Fx$. This can be modeled as the following problem:

**Problem 2.1** (Main Problem). Given a corrupted vector $y = x + e$ where $x \in \mathbb{R}^n$ is approximately $k$-sparse in the Fourier basis and $e$ is exactly $t$-sparse in the time domain (i.e. has $L_0$ norm bounded by $t$), approximate $\hat{x}_{h(k)}$.

We will solve the above problem by splitting $y$ into $x' + e' + \beta$ where $x'$ is **exactly** $k$-sparse in the Fourier domain, $e'$ is $t$-sparse in the time domain and $\beta$ is an error term bounded in $L_2$ norm by the tail of $x$. Our techniques are not limited to Fourier matrices and in fact extend naturally to other transformations like wavelets, but for simplicity we will use the term Fourier throughout.

**Related Work:** Our setting is reminiscent of extensively studied dimensionality reduction techniques like Robust PCA [CLMW11] for recovery of low rank matrices from $L_0$ corrupted data. These have wide applicability in machine learning although, in that setting, they are not able to handle truly adversarial noise and make some assumptions on the error distribution. Our results on the other hand, can protect against worst-case adversaries bounded in their $L_0$ noise budget.

Variants of the Sparse Fourier Transform have been studied [HIKP12a, HIKP12b, IKP14] but that work is concerned with recovering $\hat{x}_{h(k)}$ given an approximately sparse vector $x$, using sublinear measurements. Our focus is on recovering $\hat{x}_{h(k)}$ when some of the measurements might be corrupted and we show a tight tradeoff between the number of measurements corrupted versus the quality of recovery we can ensure.

**Our Techniques:** Our main result uses techniques from the field of compressed sensing (CS) [CRT06, BCDH10] and properties of Fourier (and related) matrices. Using these we prove that Algorithm 1 converges to a good solution to Problem 2.1, where by a good solution we mean that it is close to the true solution in the $L_\infty$ norm.

In iteration $i$ of Algorithm 1, $\hat{x}^{[i]}$ is an estimate of $\hat{x}_{h(k)}$ and $e^{[i]}$ is an estimate of $e$. In iteration $i + 1$, the algorithm uses the previous estimates, $e^{[i]}$ and $\hat{x}^{[i]}$ to update its estimates by solving the linear equation $y = F^{-1}\hat{x} + e$ and projecting onto the top $k$ Fourier coefficients of $y - e^{[i]}$. Note that while this algorithm is intuitive it does not necessarily converge to the true solution for similar settings. For example, if instead of the $L_0$ norm, $e$ was bounded in the $L_\infty$ norm, then information theoretically, there is no algorithm which can give a good solution and hence this algorithm would not be able to either. In our setting though, we can show that Algorithm 1 has an exponentially fast convergence towards a good solution to Problem 2.1 and moreover the guarantees we get are tight in the information theoretic sense. We state our result below for a general class of transformations which includes the DFT, DCT and their higher-dimensional versions.

---

**Algorithm 1** Iterative Hard Thresholding (IHT) [BCDH10].

---

    **Input:** Positive integers $k, t$, and $T$. $y = x + e$, where $x \in \mathbb{R}^n$ is approximately $k$-sparse in the
            Fourier basis and $e \in \mathbb{R}^n$ is exactly $t$-sparse in the time domain. Fourier matrix $F$.
    **Output:** $\hat{x}_{h(k)}$, approximation of the top $k$ Fourier coefficients of $x$.
1: **function** IHT($y = x + e, F, k, t, T$)
2:     $\hat{x}^{[1]} \leftarrow \vec{0}$
3:     $e^{[1]} \leftarrow \vec{0}$
4:     **for** $i = 1 \cdots T$ **do**
5:         $\hat{x}^{[i+1]} \leftarrow (F(y - e^{[i]}))_{h(k)}$
6:         $e^{[i+1]} \leftarrow (y - F^{-1}\hat{x}^{[i]})_{h(t)}$
7:     **end for**
8:     **return** $\hat{x}^{[T+1]}$
9: **end function**

---

**Theorem 2.2** (Main Theorem). *Let $F \in \mathbb{C}^{n \times n}$ be an orthonormal matrix, such that each of its entries $F_{ij}$, $|F_{ij}|$ is $O(1/\sqrt{n})$. Let $\hat{x} = Fx \in \mathbb{C}^n$ be $(k, \epsilon)$-sparse, $e \in \mathbb{R}^n$ be $t$-sparse and $y = F^{-1}\hat{x} + e$. Let $\hat{x}^{[T]} = \mathrm{IHT}(y, F, k, t, T)$, for $T = O(\log(\|x\| + \|e\|))$, then*

1. $\left\| \hat{x}^{[T]} - \hat{x}_{h(k)} \right\|_\infty = O(\sqrt{t/n} \cdot \left\| \hat{x}_{t(k)} \right\|) = O(\sqrt{t/n} \cdot \|\epsilon \hat{x}\|)$

2. $\left\| \hat{x}^{[T]} - \hat{x}_{h(k)} \right\| = O(\sqrt{kt/n} \cdot \left\| \hat{x}_{t(k)} \right\|) = O(\sqrt{kt/n} \cdot \|\epsilon \hat{x}\|)$     *(In fact (1) implies (2).)*

Note the strong $L_\infty$ - $L_2$ guarantee that Theorem 2.2 gives us, with a tight dependence between $t$, the $L_0$ budget of the adversary and $(\epsilon, k)$, the sparsity parameters of the inputs. Also, our choice of recovering just the top-$k$ coordinates of $\hat{x}$, i.e. $\hat{x}_{h(k)}$, instead of all of $\hat{x}$ is important. In the latter case, no matter what $t$ is, any solution we recover would incur an $L_2$ error of $\Omega(\left\| \hat{x}_{t(k)} \right\|)$, even when the adversary corrupts only one coordinate ($t = 1$), while in our case, with $t = o(n/k)$, we get an $L_2$ error that vanishes with $n$ and is equal to $o(\left\| \hat{x}_{t(k)} \right\|)$. (by Theorem 2.2)

### 2.2 Defending against $L_0$ budgeted adversaries

We model images as approximately $k$-sparse vectors in the 2D-DCT domain. Using the results from Section 2.1, we can recover the top-$k$ coefficients in the face of a worst-case adversary with an $L_0$ budget. To apply this to image classifiers, we want to build a neural network to recognize images projected to their top-$k$ 2D-DCT coefficients. This motivates the following framework for building classifiers that are robust to $L_0$ adversaries:

1. Train a neural network on images projected to their top-$k$ 2D-DCT coefficients. We refer to such projected images as "compressed images". [2]

2. On adversarial input images we run our $L_0$-robust DCT algorithm to recover the top-$k$ coefficients. Then transform the sparse image back to the original domain.

3. Run the recovered/corrected image through the network.

In Section 2.1, we saw that recovering the top-$k$ projection of an image gives better theoretical bounds than recovering the whole image. Hence it is important that the neural network is also trained to recognize compressed images. Training only on compressed images could possibly reduce the accuracy of neural networks, but as has been observed and used in practice (e.g. the JPEG and MPEG compression algorithms), images contain most of their information in relatively few coefficients. This is validated on our datasets, where we incur a $< 1\%$ loss in accuracy on MNIST and $< 2.5\%$ for Fashion-MNIST when training on compressed rather than original images. Note that one still needs our correction algorithm for $L_0$-corrupted images, since a naive compression of an adversarial example (by taking its top-$k$ projection) will not get classified correctly by a neural network in general. For example, if 1 pixel of the image is corrupted to have an extremely high magnitude, this would propagate into the top-$k$ coefficients of the DCT of the image too and the resulting compressed image will be nowhere close to the original uncorrupted image. Our correction algorithm does not depend

on the magnitude of the corruptions, only their number ($t$). Hence both the training of the neural network on compressed images and the correction algorithm are essential to our framework.

## 2.3 Reverse Engineering Attacks

In step 2 of our framework, we use Theorem 2.2 to get strong guarantees on the distance $\delta$, between the original compressed image $x$ and the recovered image. Ideally, $\delta$ will be so small that no adversarial examples exist in the $\delta$-ball around $x$. This may not always be achieved in practice though and there might exist a small number of adversarial examples that are in the $\delta$-ball from the original image. This leaves open the possibility, that an attacker could reverse engineer our algorithm and design an adversarial example that, when corrected, yields a (potentially different) adversarial example inside the $\delta$-ball centered at $x$ (although it is unclear how one would achieve this, as our defense is non-differentiable). Such an attack can be prevented by initializing the IHT algorithm with random vectors $\hat{x}^{[1]}, e^{[1]}$ (instead of all-zeros vectors) so that the resulting recovered image is not deterministic. Since there are only a small number of adversarial examples in the $\delta$-ball, this randomization would ensure that a reverse engineering attack would fail to hit an adversarial example, with high probability. The guarantees of the IHT algorithm (Theorem 2.2) are independent of the starting vectors and continue to hold with the randomized initialization. The IHT algorithm used for the experiments reported in this work is not randomized, because current attacks were not designed to reverse engineer our defense, and the deterministic IHT itself gives good results.

## 3 Proof of Main Result

In this section we prove Theorem 2.2, which says that Algorithm 1 converges to a good solution (one that is close to the true vector in the $L_\infty$ norm) to Problem 2.1. Our proof uses techniques from compressed sensing. The main problem studied in compressed sensing is reconstructing a signal $x$ from few linear measurements. For arbitrary signals, this task is impossible, however the main idea of compressed sensing is that signals that are approximately sparse can be recovered using fewer than $n$ linear measurements. This is modeled as,

**Problem 3.1.** Given observations $y = Mx$ where $x \in \mathbb{C}^n$ is an approximately sparse signal, and $M$ is an $m \times n$ matrix with $m < n$, recover the vector $x$.

A main success in compressed sensing (CS) is that there are efficient algorithms [BD08, NT08] for Problem 3.1 when the matrix $M$ satisfies a property called the RIP.

**Definition 3.2** (Restricted Isometry Property (RIP)). An $m \times n$ matrix $M$ has the $(k, \delta)$-*restricted isometry property* ($(k, \delta)$-RIP) if for all $k$-sparse vectors $v$ we have,

$$(1 - \delta) \cdot \|v\| \le \|Mv\| \le (1 + \delta) \cdot \|v\|.$$

Recall that in our main problem (Problem 2.1), we want to recover the top-$k$ coefficients of $\hat{x} = Fx$, where $\hat{x}$ is approximately $k$-sparse, given a corrupted vector $y = x + e$. The key idea is to notice that we can write $y$ as $\begin{bmatrix} F^{-1} & I \end{bmatrix} \begin{bmatrix} \hat{x} \\ e \end{bmatrix}$, where $\hat{x}$ is approximately $k$-sparse and $e$ is $t$-sparse. This is almost the same setup as Problem 3.1. In fact, we have more knowledge about the structure of sparsity of the vector $\begin{bmatrix} \hat{x} \\ e \end{bmatrix} \in \mathbb{C}^{2n}$ that we want to recover.

The problem of recovery with structured sparsity, has been studied under the heading of Model-Based CS [BCDH10, HIS15, HIS14, BIS17] for structured sparsity models. In our setting we want to model vectors of the form $\begin{bmatrix} \hat{x} \\ e \end{bmatrix}$, which have sparsity $k$ in $x$ and $t$ in $e$. This motivates the following definition.

**Definition 3.3.** Let $\mathcal{M}_{k,t} \subseteq \mathbb{C}^{2n}$ be the set of all vectors where the first $n$ coordinates are $k$-sparse and the last $n$ coordinates are $t$-sparse.[3] Formally,

$$\mathcal{M}_{k,t} := \{v = \begin{bmatrix} x \\ e \end{bmatrix} \in \mathbb{C}^{2n} \mid x \text{ is } k\text{-}sparse, e \text{ is } t\text{-sparse}\}.$$

We say that a matrix $M$ has the $((k, t), \delta)$-RIP if for all vectors $v \in \mathcal{M}_{k,t}$,

$$(1 - \delta) \cdot \|v\| \leq \|Mv\| \leq (1 + \delta) \cdot \|v\|.$$

Model-Based CS was first introduced in [BCDH10], for general sparsity models, and they proved therein that Iterative Hard Thresholding (IHT) [BD08] indeed converges to a good solution to Problem 3.1, given that the measurement matrix $M$ satisfies RIP for the model. We use this Model-Based IHT approach to argue that Algorithm 1 finds a good solution to Problem 2.1. In [BCDH10], they proved that the IHT algorithm converges to an approximately correct solution, given that the measurement matrix $M$ satisfies the RIP for the model at hand. For us this translates to the following theorem.

**Theorem 3.4** ([BCDH10]). *Let $v \in \mathcal{M}_{k,t}$ and let $y = Mv + \beta$, where $M \in \mathbb{R}^n$ is a full-rank matrix and $\beta$ is a noise vector. Let $v^{[T]} = \text{IHT}(y, M^{-1}, k, t, T)$. If $M$ is $((3k, 3t), \delta)$-RIP, with $\delta \leq 0.1$, then*

$$\left\| v^{[T]} - v \right\| \leq 2^{-T} \cdot \|v\| + 4 \cdot \|\beta\|.$$

We use the above theorem to prove that Algorithm 1 also converges to a good solution. Another key technique we use in our proofs is an uncertainty principle for specific structured matrices.

**Lemma 3.5** (General Uncertainty Principle). *Let $F$ be a matrix in $\mathbb{C}^{n \times n}$ such that each entry $F_{ij}$ has $|F_{ij}| \leq \alpha$. Let $x$ be a $k$-sparse vector in $\mathbb{C}^n$ and $y = Fx$. Then $\|y\|_\infty \leq \alpha \cdot \sqrt{k} \cdot \|x\|$.*

Note that when $F$ is the normalized Fourier matrix, this is the same as the folklore Fourier uncertainty principle with $\alpha = 1/\sqrt{n}$. The proof of the above is indeed very similar to the Fourier case and is included in the full version of the paper[4]. One can check that for transformation matrices corresponding to Discrete Cosine and Sine Transforms and their 2D analogs we have $\alpha = O(\sqrt{1/n})$.

Finally to prove Theorem 2.2, we first prove that the matrix $M = [F^{-1} \ I]$ has the RIP (Lemma 3.6 below), which then combined with Theorem 3.4 and the uncertainty principle 3.5, finishes the proof of Theorem 2.2. These proofs can be found in the full version of the paper.

**Lemma 3.6.** *Let $F$ be an orthonormal matrix, such that each entry $F_{ij}$ has $|F_{ij}| = O(1/\sqrt{n})$. Then the matrix $M = [F^{-1} \ I] \in \mathbb{C}^{n \times 2n}$ satisfies $((3k, 3t), \delta)$- RIP with $\delta \leq 0.1$, when $t = O(n/k)$. Equivalently, for all vectors $v = \begin{bmatrix} \hat{x} \\ e \end{bmatrix}$, such that $\hat{x}$ is $3k$-sparse and $e$ is at most $3t = O(n/k)$-sparse,*

$$(1 - \delta) \cdot \|v\| \leq \|Mv\| \leq (1 + \delta) \cdot \|v\|.$$

## 4 Experiments

### 4.1 Worst case adversaries

We evaluated our framework on three leading $L_0$ attacks in the literature: the JSMA Attack of Papernot et al [PMJ+15], the $L_0$ attack from Carlini and Wagner (CW) [CW16], and the adversarial patch from Brown et al [BMR+17]. We evaluated Algorithm 1 on the JSMA and CW attacks and present these results in this section. We discuss experiments on the adversarial patch in Section 4.2.

We tested both JSMA and CW on two datasets: the MNIST handwritten digits [LeC98] and the Fashion-MNIST [XRV17] dataset of clothing images. For each attack, we used randomly selected targets. For both datasets we used a neural network composed of a convolutional layer (32 kernels of 3x3), max pooling layer (2x2), convolutional layer (64 kernels of 3x3), max pooling layer (2x2), fully connected layer (128 neurons) with dropout (rate = .25) and an output softmax layer (10 neurons). We used the Adam optimizer with cross-entropy loss and ran it for 10 epochs over the training datasets.

For each dataset, we trained our neural network only on images that were projected onto their top-$k$ 2D-DCT coefficients. Here $k$ is a parameter we tuned depending on the dataset (for MNIST $k = 40$ and for Fashion-MNIST $k = 35$). For each dataset, we fixed its corresponding $k$ across all experiments reported here.

In all of our evaluations there were three experimental conditions: first we ran uncorrupted images through the network to establish a baseline accuracy. Then we ran the $L_0$ adversarial examples through the network. Finally, we ran our correction algorithm on the adversarial examples and ran the results through the network. Example images of these conditions can be seen in Figure 1.

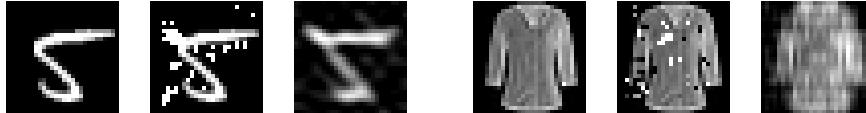

Figure 1: Example experimental conditions. The left 3 images depict an original MNIST image, the image corrupted by JSMA, and our corrected image. The right three images show an original Fashion-MNIST image, the image corrupted by CW, and our corrected image.

For the JSMA, we ran an experiment for several different adversary noise budgets. For each budget, we evaluated the network on the three experimental conditions. The accuracy vs $L_0$ budget and loss vs $L_0$ budget graphs can be seen in Figure 3 on the MNIST and Fashion-MNIST datasets. Exact values can be found in the full version of the paper. The results demonstrate that our correction algorithm successfully defends against the JSMA attack. For example when the adversary corrupts 30 bits, it is able to drop the accuracy of our network on the Fashion-MNIST dataset from $88.5\%$ to $24.8\%$ but after running our recovery algorithm we get back up to $83.1\%$.

The CW attack works by finding a minimal set of pixels that can be corrupted to fool the network. This means that the adversary's budget will depend on the particular image being corrupted rather than being fixed in advance. For this reason, we let the CW adversary choose how many pixels to corrupt and allow ourselves to know its budget for each image. Note that the locations and magnitudes of the noise are unknown to us. Since the budget varies across images, a plot like Figure 3 does not make sense and we instead report the overall accuracy and loss of our correction algorithm in Table 2. Again our correction algorithm is effective against CW. For example on the Fashion-MNIST dataset the network's test accuracy on original images was $87.8\%$. The CW attack was successful and the network mislabeled every adversarial example. After running our correction algorithm, the accuracy returns to $85.7\%$.

|  | MNIST | Adversarial | Corrected | F-MNIST | Adversarial | Corrected |
|---|---|---|---|---|---|---|
| Accuracy | 99.0 | 0.0 | 72.8 | 87.8 | 0.0 | 85.7 |
| Loss | 0.002 | 0.115 | 0.095 | 0.035 | 0.140 | 0.040 |

Figure 2: Experimental results for our algorithm on the CW attack for the MNIST and Fashion-MNIST datasets. Columns 2-4 show results for MNIST data and columns 5 to 7 show Fashion-MNIST.

As images grow larger they become less sparse in Fourier bases but natural images are still block-wise sparse. In such cases our algorithm could be modified to correct images block by block, in which case the network would need to be trained on images compressed block by block (e.g. as in JPEG). This would work with the mild assumption that the corrupted locations are well-distributed across blocks because then our recovery result could be applied to each block separately. Within each block the corrupted locations could still be anywhere and of any magnitude. For images that are too large to be sparse in Fourier bases, the block-wise approach may fail in the case where most of the $L_0$ noise resides in few blocks because in these blocks there will be too many corrupted coordinates to recover. In the next section we study the extreme case where all of the error is concentrated contiguously. We show that even in this extreme case our framework for $L_0$-robust sparse transformations can be used to guard against contiguous noise attacks even in large images.

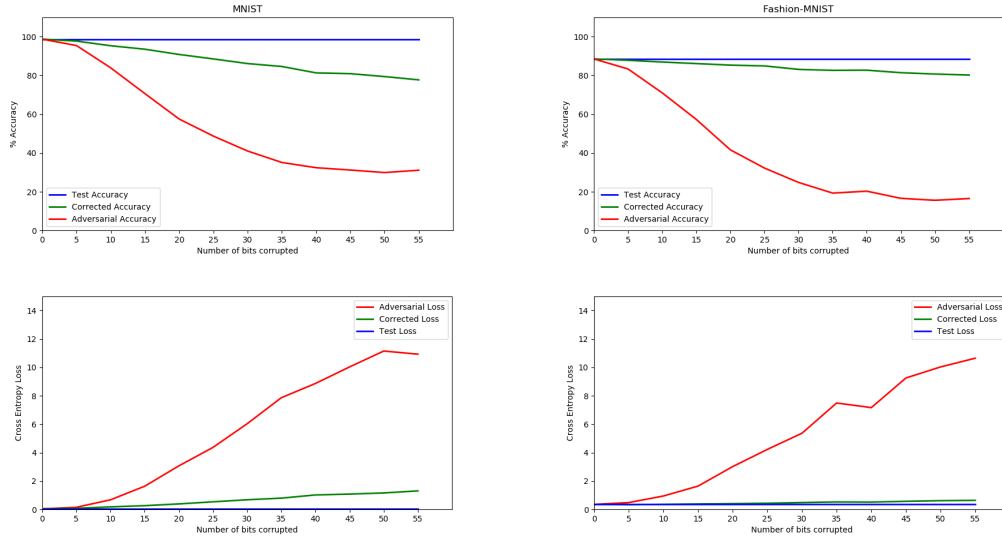

Figure 3: Classification accuracy and loss for JSMA on MNIST (left) and Fashion-MNIST (right). Blue lines show the performance of the network on original images (and hence does not change with the number of coordinates corrupted). Red lines show the performance of the network on uncorrected adversarial examples and green lines show the performance of the network on images that were corrected by Algorithm 1

## 4.2 Adversarial patch

In [BMR$^+$17], the authors introduce a method for generating adversarial patches. These are targeted attacks in the form of circular images that get overlayed on input images. They showed that their patch effectively fools leading image classifiers into mislabeling patched images.

Notice that the adversarial patch is an example of $L_0$ noise and so fits within our framework. The patch attack is only successful when the patch is sufficiently large (~ 80 pixels in diameter for $224 \times 224$ images), which is larger than our algorithm can tolerate. Also images of this size are less sparse in the frequency domain and as discussed above, our approach may not be able to correct contiguous noise on such images. Similarly we cannot train the neural networks on compressed images as that would lead to non trivial loss as the images are less sparse. So in this section we use a network that was pretrained on original ImageNet images.

We are able to use the contiguity of the noise with the mild sparsity of large images by using the Patchwise IHT Algorithm to defend against the patch attack. Since image recovery is not possible in this setting, our algorithm instead focuses on *detecting* the location of the contiguous noise. We detect the noise by searching over contiguous blocks in the image and running Algorithm 1 on each block, where we project $e$ only to the block rather than top-$t$ coordinates. Finally we find the block for which the remaining image $(y - e)$ is sparsest in the Fourier domain. We call this Patchwise IHT and a formal description of the algorithm is given in the full version of the paper. Note that for this particular set of adversarial examples there may be other ways to detect the patch with pre-processing. We do not do any such optimizations that are particular to the adversary and Patchwise IHT is based only on the mild sparsity of the original images.

We took 700 random images from ImageNet and for classification we used pretrained ResNet-50 network [HZRS15]. We ran each image through the network in our three experimental conditions, depicted in Figure 4.

Figure 5 shows the results of our experiment. The patch was a successful attack (Top-5 accuracy dropped from $92.3\%$ to $63.9\%$ and Top-1 from $76.4\%$ to $12.0\%$). After correcting, Top-5 accuracy jumped to $80.4\%$ (Top-1: $59.7\%$). Only $1.0\%$ of the original images were labeled as 'toaster' (none in the Top-1), but 'toaster' was in the Top-5 in $99.0\%$ of the patched images with $85.7\%$ being the

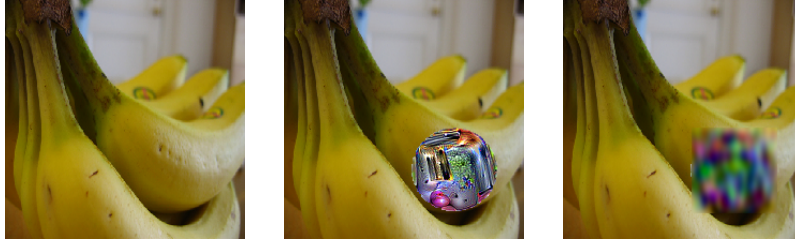

Figure 4: Example of the three image conditions in the patch experiment. Left is the original image classified as 'banana' with probability $.94$. The middle, with the adversarial patch overlayed, is classified as 'toaster' with probability $.93$. The right is the image after our Patchwise IHT algorithm, which gets classified as 'banana' with probability $.94$.

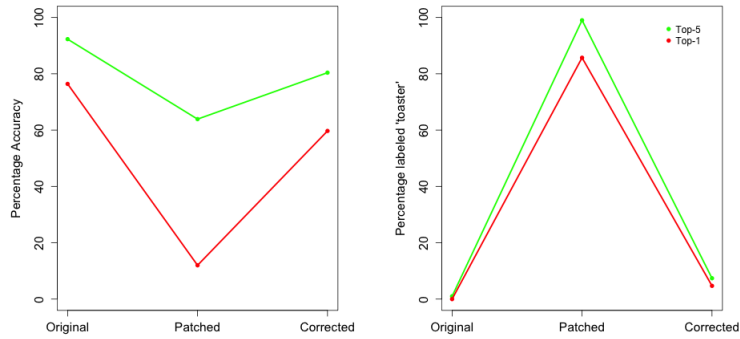

Figure 5: Experimental results for Patchwise IHT algorithm. The left plot depicts the accuracy of the network in our three experimental conditions. The right plot shows the percentage of images labeled as 'toaster' under the same three conditions.

most confident label. Notably, very few corrected images were labeled as 'toaster' (Top-5: $7.4\%$, Top-1: $4.7\%$).

# 5   Acknowledgements

Mitali Bafna was supported by NSF Grant CCF 1715187. Jack Murtagh was supported by NSF grant CNS-1565387. Nikhil Vyas was supported by an Akamai Presidential Fellowship and NSF Grant CCF-1552651. We would like to thank Yaron Singer and Adam Breuer for helpful feedback and encouragement in the early stages of this work. We also want to thank Thibaut Horel for valuable comments on the manuscript. Thanks also to the reviewers for helpful remarks.

## Footnotes

[2]Indeed the JPEG lossy-compression algorithm essentially does such a top-$k$ projection!

[3]Recall that $\mathcal{M}_k$ was the set of all $k$-sparse vectors in $\mathbb{C}$. Note that $\mathcal{M}_{k,t}$ is different from $\mathcal{M}_{k+t}$ which is the set of all vectors $\in \mathbb{C}^{2n}$ that are $k + t$-sparse.

[4]https://arxiv.org/pdf/1812.05013.pdf

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
