[Supplementary Material]

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

## A  Proof of Lemma 3.5: the General Uncertainty Principle

*Proof.* For all $i \in [n]$ we have

$$
\begin{aligned}
|(Fx)_i| &= \left| \sum_{j \in [n]} F_{ij} x_j \right| \\
&\leq \sum_{j \in [n]} |F_{ij}| \cdot |x_j| \\
&\leq \alpha \cdot \sum_{j \in [n]} |x_j| \\
&\leq \alpha \cdot \sqrt{k} \cdot \|x\|
\end{aligned}
$$

where the third line follows from our assumption on the entries in $F$ and the fourth line follows from the Cauchy-Schwarz inequality. Since $y = Fx$ we have $\|y\|_\infty = \|Fx\|_\infty = \max_{i \in [n]}(|(Fx)_i|) \leq \alpha \cdot \sqrt{k} \cdot \|x\|$ by the above. $\qquad\square$

## B  Proof of Lemma 3.6

*Proof.* We will prove that $[F^{-1}\ I]$ has the RIP for all $v = \begin{bmatrix} \hat{x} \\ e \end{bmatrix} \in \mathcal{M}_{3k,3t}$. That is,

$$
0.9 \cdot \left\| \begin{bmatrix} \hat{x} \\ e \end{bmatrix} \right\| \leq \left\| [F^{-1}\ I] \begin{bmatrix} \hat{x} \\ e \end{bmatrix} \right\| \leq 1.1 \cdot \left\| \begin{bmatrix} \hat{x} \\ e \end{bmatrix} \right\|.
$$

Note that the right-hand inequality follows immediately: Since $F^{-1}$ is orthonormal we have that
$\left\| [F^{-1}\ I] \begin{bmatrix} \hat{x} \\ e \end{bmatrix} \right\| = \|F^{-1}\hat{x} + e\| \leq \|F^{-1}\hat{x}\| + \|e\| = \|\hat{x}\| + \|e\| = \left\| \begin{bmatrix} \hat{x} \\ e \end{bmatrix} \right\| \leq 1.1 \left\| \begin{bmatrix} \hat{x} \\ e \end{bmatrix} \right\|.$

Since $x$ is $k$-sparse and $F$ satisfies the hypotheses of the Uncertainty Principle (Lemma 3.5), with $\alpha = O(1/\sqrt{n})$, we have that $\|F^{-1}\hat{x}\|_\infty \leq \alpha \cdot \sqrt{k} \cdot \|\hat{x}\|$. Using this, we will prove that, $\|F^{-1}\hat{x} + e\| \geq 0.9\sqrt{\|\hat{x}\|^2 + \|e\|^2}$, when the sparsity of $e$ is at most $O(n/k)$.

Since $e$ is $t$-sparse, without loss of generality assume that $e = [e_1, \ldots, e_t, 0, \ldots, 0]$ and $F^{-1}\hat{x} = [x_1, \ldots, x_n]$ with $|x_i| \leq \alpha \cdot \sqrt{k} \cdot \|x\|$. We have that,

$$
\begin{aligned}
\left\| F^{-1}\hat{x} + e \right\|^2 &\geq \sum_{i=1}^{t} (|x_i| - |e_i|)^2 + \sum_{i=t+1}^{n} x_{i+1}^2 \\
&= \|x\|^2 + \|e\|^2 - 2\sum_{i=1}^{t} |x_i| \cdot |e_i| \\
&\geq \|x\|^2 + \|e\|^2 - 2 \cdot \alpha \cdot \sqrt{k} \|x\| \sum |e_i| \qquad &\text{(Uncertainty Principle)} \\
&\geq \|x\|^2 + \|e\|^2 - 2 \cdot \alpha \cdot \sqrt{k} \|x\| \cdot \sqrt{t} \|e\| \qquad &\text{(Cauchy-Shwartz inequality)} \\
&= \|x\|^2 + \|e\|^2 - O(\sqrt{kt/n} \|x\| \|e\|) \qquad &\text{(1)}
\end{aligned}
$$

We want that $\|F^{-1}x + e\| \geq 0.9\sqrt{\|x\|^2 + \|e\|^2}$. Plugging in equation 1 and moving terms around we get that this happens when $t = O(n/k)$. This completes the proof of the lemma. $\qquad\square$

## C  Proof of Theorem 2.2

*Proof of the main theorem:* Now we will prove that the RIP property of $M = [F^{-1}\ I]$ proved above, combined with the Uncertainty Principle (Lemma 3.5) and Theorem 3.4 imply the main theorem.

Consider $y = F^{-1}\hat{x} + e = F^{-1}\hat{x}_{h(k)} + F^{-1}\hat{x}_{t(k)} + e$, where $\|\hat{x}_{t(k)}\| \leq \epsilon \|x\|$ and $e$ is $t$-sparse. Using $\beta = F^{-1}\hat{x}_{t(k)}$, we can rewrite this expression as, $y = [F^{-1} \ I] \begin{bmatrix} \hat{x}_{h(k)} \\ e \end{bmatrix} + \beta = Mv + \beta$, where $\|\beta\| = \|F^{-1}\hat{x}_{t(k)}\| = \|\hat{x}_{t(k)}\| \leq \epsilon \|x\|$, since $F$ is orthonormal. In Lemma 3.6 we proved that the matrix $M$ is $((3k, 3t), \delta)$-RIP (with $\delta = 0.1$) for the set $\mathcal{M}_{k,t}$.

At the $i^{th}$ iteration of the IHT algorithm let $\begin{bmatrix} \hat{x}^{[i]} \\ e^{[i]} \end{bmatrix}$ be our estimate of $\begin{bmatrix} \hat{x}_{h(k)} \\ e \end{bmatrix}$. At the $T^{th}$ iteration, by Theorem 3.4 we have the guarantee that,

$$\left\| \begin{bmatrix} \hat{x}^{[T]} \\ e^{[T]} \end{bmatrix} - \begin{bmatrix} \hat{x}_{h(k)} \\ e \end{bmatrix} \right\| \leq 2^{-T} \cdot \sqrt{\|x\|^2 + \|e\|^2} + 4 \|\beta\| \approx 4\epsilon \|\hat{x}\| \tag{2}$$

$$\implies \left\|\hat{x}^{[T]} - \hat{x}_k\right\|^2 + \left\|e^{[T]} - e\right\|^2 \leq 16\epsilon^2 \|\hat{x}\|^2, \tag{3}$$

since we set $T$ such that $2^{-T} \cdot \sqrt{\|x\|^2 + \|e\|^2} \approx 0$. Note that 3 already gives a weak $L_2$-$L_2$ guarantee on $\|\hat{x}^{[T]} - \hat{x}_{h(k)}\|$ but we will derive a stronger $L_\infty$-$L_2$ guarantee.

Consider the $T^{th}$ iteration of IHT, and define a vector $z := F(y - e^{[i-1]})$. The IHT algorithm sets $\hat{x}^{[T]} = z_{h(k)}$. We have that,

$$y = F^{-1}\hat{x} + e = F^{-1}z + e^{[i-1]} \Leftrightarrow \hat{x} - z = F(e^{[T-1]} - e)$$

Since both $e, e^{[T-1]}$ are $t$-sparse, we have that the vector $e^{[T-1]} - e$ is $2t$-sparse. By the uncertainty principle 3.5, we get that, $\|\hat{x} - z\|_\infty \leq \sqrt{2t/n} \|e^{[T]} - e\|_2 = O(\epsilon \|\hat{x}\| \sqrt{t/n})$ by Equation 3. This trivially implies that $\|\hat{x}_{h(k)} - z_{h(k)}\|_\infty = O(\epsilon \|\hat{x}\| \sqrt{t/n})$ which implies $\|\hat{x}_{h(k)} - z_{h(k)}\| = O(\epsilon \|\hat{x}\| \sqrt{kt/n})$. $\qquad \square$

## D  JSMA experimental results

| MNIST Accuracy | 0 | 5 | 10 | 15 | 20 | 25 |
|---|---|---|---|---|---|---|
| Original Images | 98.7 | 98.7 | 98.7 | 98.7 | 98.7 | 98.7 |
| Adversarial Images | 98.7 | 95.4 | 83.9 | 70.6 | 57.5 | 48.7 |
| Corrected Images | 98.7 | 97.7 | 95.3 | 93.5 | 90.8 | 88.5 |

| MNIST Accuracy | 30 | 35 | 40 | 45 | 50 | 55 |
|---|---|---|---|---|---|---|
| Original Images | 98.7 | 98.7 | 98.7 | 98.7 | 98.7 | 98.7 |
| Adversarial Images | 41.0 | 35.1 | 32.4 | 31.2 | 29.9 | 31.1 |
| Corrected Images | 86.1 | 84.6 | 81.3 | 80.9 | 79.4 | 77.7 |

| MNIST Loss | 0 | 5 | 10 | 15 | 20 | 25 |
|---|---|---|---|---|---|---|
| Original Images | 0.05 | 0.05 | 0.05 | 0.05 | 0.05 | 0.05 |
| Adversarial Images | 0.05 | 0.16 | 0.68 | 1.64 | 3.07 | 4.37 |
| Corrected Images | 0.05 | 0.08 | 0.19 | 0.27 | 0.39 | 0.54 |

| MNIST Loss | 30 | 35 | 40 | 45 | 50 | 55 |
|---|---|---|---|---|---|---|
| Original Images | 0.05 | 0.05 | 0.05 | 0.05 | 0.05 | 0.05 |
| Adversarial Images | 6.03 | 7.86 | 8.86 | 10.03 | 11.15 | 10.92 |
| Corrected Images | 0.69 | 0.80 | 1.02 | 1.09 | 1.17 | 1.31 |

Figure 6: Experimental results for the JSMA attack on MNIST. The columns represent the adversary's budget: the number of pixels corrupted from 0 to 55 in increments of 5, given in the first row. The top two tables show accuracy results on the MNIST for the three experimental conditions. The bottom two tables show the cross-entropy loss across the the conditions and budgets.

| F-MNIST Accuracy | 0 | 5 | 10 | 15 | 20 | 25 |
|---|---|---|---|---|---|---|
| Original Images | 88.5 | 88.5 | 88.5 | 88.5 | 88.5 | 88.5 |
| Adversarial Images | 88.5 | 83.3 | 71.0 | 57.3 | 41.6 | 32.2 |
| Corrected Images | 88.5 | 87.8 | 86.9 | 86.1 | 85.3 | 84.9 |

| F-MNIST Accuracy | 30 | 35 | 40 | 45 | 50 | 55 |
|---|---|---|---|---|---|---|
| Original Images | 88.5 | 88.5 | 88.5 | 88.5 | 88.5 | 88.5 |
| Adversarial Images | 24.8 | 19.3 | 20.3 | 16.6 | 15.6 | 16.5 |
| Corrected Images | 83.1 | 82.6 | 82.7 | 81.4 | 80.7 | 80.2 |

| F-MNIST Loss | 0 | 5 | 10 | 15 | 20 | 25 |
|---|---|---|---|---|---|---|
| Original Images | 0.36 | 0.36 | 0.36 | 0.36 | 0.36 | 0.36 |
| Adversarial Images | 0.36 | 0.49 | 0.95 | 1.65 | 3.02 | 4.23 |
| Corrected Images | 0.36 | 0.34 | 0.37 | 0.39 | 0.41 | 0.44 |

| F-MNIST Loss | 30 | 35 | 40 | 45 | 50 | 55 |
|---|---|---|---|---|---|---|
| Original Images | 0.36 | 0.36 | 0.36 | 0.36 | 0.36 | 0.36 |
| Adversarial Images | 5.37 | 7.49 | 7.17 | 9.25 | 10.03 | 10.64 |
| Corrected Images | 0.49 | 0.53 | 0.52 | 0.58 | 0.62 | 0.65 |

Figure 7: Experimental results for the JSMA attack on Fashion-MNIST. The columns represent the adversary's budget: the number of pixels corrupted from 0 to 55 in increments of 5, given in the first row. The top two tables show accuracy results on the Fashion-MNIST for the three experimental conditions. The bottom two tables show the cross-entropy loss across the the conditions and budgets.

# E   Patchwise Iterative Hard Thresholding

---
**Algorithm 2** Patchwise Iterative Hard Thresholding (IHT)
---
**Input:** Positive integers $k, t, T$, and $\ell$. $y = x + e$, where $x \in \mathbb{R}^n$ is approximately $k$-sparse in the Fourier basis and $e \in \mathbb{R}^n$ is exactly $t$-sparse in the time domain. Fourier matrix $F$.
**Output:** $x'$, approximation of the original signal $x$.

1: **function** IHT($y = F^{-1}\hat{x} + e, F, k, t, T, \ell$)
2:     $x' \leftarrow \vec{\infty}$
3:     **for** $\ell \times \ell$ patch $p$ in image **do**
4:         $\hat{x}^{[1]} \leftarrow \vec{0}$
5:         $e^{[1]} \leftarrow \vec{0}$
6:         **for** $i = 1 \cdots T$ **do**
7:             $\hat{x}^{[i+1]} \leftarrow (F(y - e^{[i]}))_{h(k)}$
8:             $e^{[i+1]} \leftarrow (y - F^{-1}\hat{x}^{[i]})_{h(t)}$
9:         **end for**
10:        **if** $\left\| \hat{x}^{[T+1]} \right\| < \|x'\|$ **then**
11:            $x' = F^{-1}\hat{x}^{[T+1]}$
12:        **end if**
13:    **end for**
14:    **return** $x'$
15: **end function**
---