[Reviews · NeurIPS 2018]

Reviewer 1



The authors give a framework for improving the robustness of classifiers to adversaries, which is based on a Sparse Discrete Fourier transformation that is robust to worst-case L0 noise. The techniques, as you suggest, can correct the corruptions made by the adversarial attacks and produce an approximate image which is close to the original image. However, I have some concerns that need further clarification from the authors: 1.     Since your approach owns a benefit of generality, i.e., the framework is a general purpose, is it possible to attacks your algorithm in a targeted manner? In other words, if we input the same image twice, we can get the same approximation results in certain parameters setting, right? Thus, does it mean that your algorithm is deterministic? Is this the drawback that can be easily found or contradicted? For example, as shown in Figure 4, can we design an attack image as the input of Patchwise IHT algorithm, yet, output an image as exact as figure 4(b). 2.     In Section 4 Experiments, have you compared with other defended model e.g. against the L0 or L1 noise? If no, please perform it and make a comparison.

Reviewer 2



The authors present an algorithm that can defend against worst-case adversarial attacks on image classification, for adversaries that are allowed to affect a maximum number of pixels in the input image. They also provide theoretical guarantees for the method, as well as good motivation with practical uses (I particularly liked the comment about compression algorithms like JPEG and MPEG making the same assumptions). Finally, the presented experimental results are compelling. It would be good to add details on how the models were trained (i.e., optimizer and convergence criteria), so that the results are reproducible. Some minor typos/comments: - Line 9: What is “CW”? Noticed later that it refers to the names of the authors. You should probably make that explicit in the abstract as it is currently confusing. - Line 65: “In section 4 we” -> “In section 4, we” - Line 72: “In section 4 we” -> “In section 4, we” - Line 74: Missing a space before the last opening square bracket. - Line 86: “all zeros” -> “all-zeroes” - Line 98: “In other words if” -> “In other words, if” - Line 109: Missing a space before the opening square bracket. - Line 110: “although in that” -> “although, in that” - Line 111: “and have some” -> “and make some” - Line 113: “but their work” -> “but that work” - Algorithm 1: It may help to define F, k, t, and T, in the algorithm definition. - Line 138: “Also our” -> “Also, our” - Line 154: “In section 2.1 we” -> “In section 2.1, we” - Line 177: Missing space before the opening parenthesis and the opening square bracket. - Line 184: “In fact we” -> “In fact, we” - Line 191: Missing space before the opening square bracket.

Reviewer 3



The authors use tools from compressive sensing to defend classifiers against adversarial corruption of images with L0 constrained attacks. Specifically, assuming the input image is approximately sparse in the Fourier domain, they can approximately recover the adversarial perturbations and the large Fourier coefficients of the input image. They test their defense method on known L0 attacks and they are able to defend against these attacks. I found their defense algorithm to be a clever use of compressive sensing. They show that the observed signal y is a matrix transform of the input and errors. They then show that this particular matrix satisfies the RIP condition for the input model of interest. Typically we show the RIP condition of random low-rank matrices. It was interesting to exploit the RIP condition of this particular deterministic and relatively high-rank matrix. My issue with this work is that the theoretical results do not precisely imply defense against L0 attacks. It only shows that we can approximately recover the large Fourier coefficients of the image. While the proposed algorithm does defend against known attacks, perhaps with knowledge of the defense method it may be possible to still attack the classifier. For example in Figure 4 there are some large artifacts after the recovery. An attack may be able to exploit the artifacts left over after running the recovery algorithm. Overall, I believe that this is a good contribution to the area with some nice ideas. However there is still work to be done and I would not be surprised if an attack is developed that can bypass this defense. *** Post Rebuttal *** Thank you for your response. Since this work only allows approximate recovery in L_inf, and we know that there exists adversarial examples in L_inf, I still think that an attacker with knowledge of the defense can still cause the network to output the wrong result. I still think that this work makes a nice contribution to the space.